# Numerical Simulation and Mechanical Properties of 6063/6082 Dissimilar Joints by Laser Welding

Shuwan Cui [1,2,3,*], Fuyuan Tian [2], Suojun Zhang [3], Hongfeng Cai [3] and Yunhe Yu [2]

1   HNU College of Mechanical and Vehicle Engineering, Hunan Univerisity, Changsha 410082, China
2   School of Mechanical and Automotive Engineering, Guangxi University of Science and Technology, Liuzhou 545006, China; 221068248@stdmail.gxust.edu.cn (F.T.); 221068163@stdmail.gxust.edu.cn (Y.Y.)
3   Dongfeng Liuzhou Automobile Co., Ltd., Liuzhou 545006, China; 19977435329@163.com (S.Z.); 13233595424@163.com (H.C.)
*   Correspondence: swcui@gxust.edu.cn; Tel.: +86-135-9706-6615

**Abstract:** In this paper, the laser welding process of 6082-T6 and 6063-T6 dissimilar aluminum alloys with a thickness of 2.5 mm was numerically simulated by using a rotary surface Gauss heat source and the flow state of the weld pool was analyzed. The microstructure and mechanical properties of the welded joint (WJ) with a laser power of 1.75 kW were also studied. The results show that the recoil pressure in the molten pool tends to be stable with the increase in welding power, and the surface tension was the main driving force affecting the liquid metal flow in the molten pool. Under the action of 1.75 kW of laser power, the macromorphology of the weld was complete, continuous, and clear. The weld metal zone (WMZ) near both sides of the fusion line (FL) was columnar in microstructure, and the center of the WMZ was dominated by equiaxed crystals. The average microhardness of WMZ was 73.46 HV, which was lower than the base material zone (BM) and heat-affected zone (HAZ). The fracture region of the tensile specimen was located in HAZ on the 6063-T6 side of WJ, showing ductile fracture characteristics with a tensile strength of 180.8 MPa and elongation of 4.04%.

**Keywords:** numerical simulation; dissimilar welded joint; mechanical properties; microstructure

## 1. Introduction

According to the International Aluminum Association (IEA), the share of aluminum was 70,248 thousand metric tonnes in the past year, accounting for 10% of the global industrial metals market. The 6-series aluminum alloys accounted for about 10% of the global total aluminum. The 6063-T6 [1] and 6082-T6 [2] were the most used series of the 6-series aluminum alloys, with a share of nearly 70% and a total share of about 4917.36 thousand metric tonnes. These alloys, characterized by their low density (approximately one-third of steel), high strength, excellent corrosion resistance, and superior plastic formability, have become increasingly prevalent in the automotive manufacturing sector. Table 1 presents a compilation of commonly used material property data. In an effort to reduce production costs and fulfill the performance criteria for various automobile components, the employment of dissimilar aluminum alloy welding processes for assembling different parts of vehicles is becoming indispensable. The integrity of the entire structure hinges significantly on the welding quality. Aluminum alloys, however, often encounter numerous welding challenges due to their high thermal conductivity, rapid solidification rate, broad solidification temperature range, and thermal expansion coefficient, leading to issues such as porosity, hot cracking, and softening. Thus, achieving high-quality welding of dissimilar aluminum alloys is not only of great scientific importance but also holds substantial practical value in enhancing automotive safety performance and advancing the lightweight automotive industry.



**Table 1.** Common parameters for materials.

| Materials | Density (g/cm$^3$) | Shear Strength (MPa) | Tensile Strength (MPa) | Elongation at Break (%) | Thermal Diffusivity (mm$^2$/s) | Melting Point (°C) |
|---|---|---|---|---|---|---|
| 6063-T6 | 2.7 | 150 | 260 | 8.0 | 83 | 620 |
| 6082-T6 | 2.7 | 220 | 330 | 9.8 | 67 | 580 |

Tungsten inert gas (TIG) welding, metal inert gas (MIG) welding, and friction stir welding are traditional aluminum alloy welding methods. These conventional techniques often result in defects such as collapse, undercut, pores, and thermal cracks when applied to 6-series aluminum alloys, leading to diminished mechanical properties and reduced corrosion resistance of the WJs. Laser welding, characterized by its high energy density and minimal wear consumption (such as electrode absence), offers minimal thermal deformation, making it particularly apt for welding thin aluminum alloys [3]. This method has thus garnered significant attention in the field of aluminum alloy welding. Chu et al. [4] investigated the impact of varied welding heat inputs on the structure, texture, and mechanical properties of 6061 aluminum alloy laser WJs. Findings revealed that the welds predominantly comprised columnar and equiaxed grains, with coarse columnar dendrites exhibiting a pronounced cubic texture ({001} <100>), resulting in a marked decrease in welding strength. The tensile specimens consistently displayed ductile fractures at the weld, with softening in the heat-affected zone (HAZ) due to recrystallization. After laser welding 6082-T6 and A357 thin sheets, D.V. et al. [5] investigated the impact of welding parameters on the weld formation. It was discovered through mechanical property testing and microstructure observation that post-heat treatment had the biggest impact on the joint's ultimate performance and that faster welding speeds decreased the porosity in the weld pass. For the purpose of circular oscillation laser welding 6061 aluminum alloy, Ai et al. [6] developed a three-dimensional numerical simulation model as well as an energy distribution model. They also investigated the impact of weld morphology on the quality of welded joints. The peak values of the energy density and weld width decreased as the oscillation frequency increased. This finding, which clarifies the mechanism underlying the weld width difference, is crucial for raising the caliber of oscillating laser welding aluminum alloy. Leo et al. [7] analyzed how welding parameters influence the quality of Al-Mg alloy laser welding, concluding that porosity and Mg content in the fusion zone significantly affect tensile strength and elongation. A defined power distribution was suggested to stabilize the welding process and minimize weld porosity. In summary, extensive research has been conducted on aluminum alloy laser welding technology to enhance the quality of laser WJs and optimize their microstructure and properties. However, current research predominantly focuses on similar aluminum alloys, with a notable lack of studies on the microstructure and properties of laser WJs of dissimilar aluminum alloys.

This paper delves into the microstructure evolution and mechanical properties of laser WJs, coupled with a numerical simulation of the flow field during the laser welding process. The study's insights provide a theoretical framework for applying dissimilar aluminum alloy welding in relevant industries.

## 2. Materials and Methods

### 2.1. Experimental Parameters and Materials

Figure 1 illustrates the setup of the laser welding process. For shielding, Argon gas (99.99%) was used at a flow rate of 20 L/min (WFF3000, Dazhu, Shenzhen, China). The welding speed was maintained at 10 mm/s, while laser power settings were adjusted to 1.50 kW, 1.75 kW, and 2.00 kW. The experiment employed 6063-T6 and 6082-T6 aluminum alloys, each with a thickness of 2.5 mm, as the base materials (BM). Table 2 details the chemical composition of the BM.

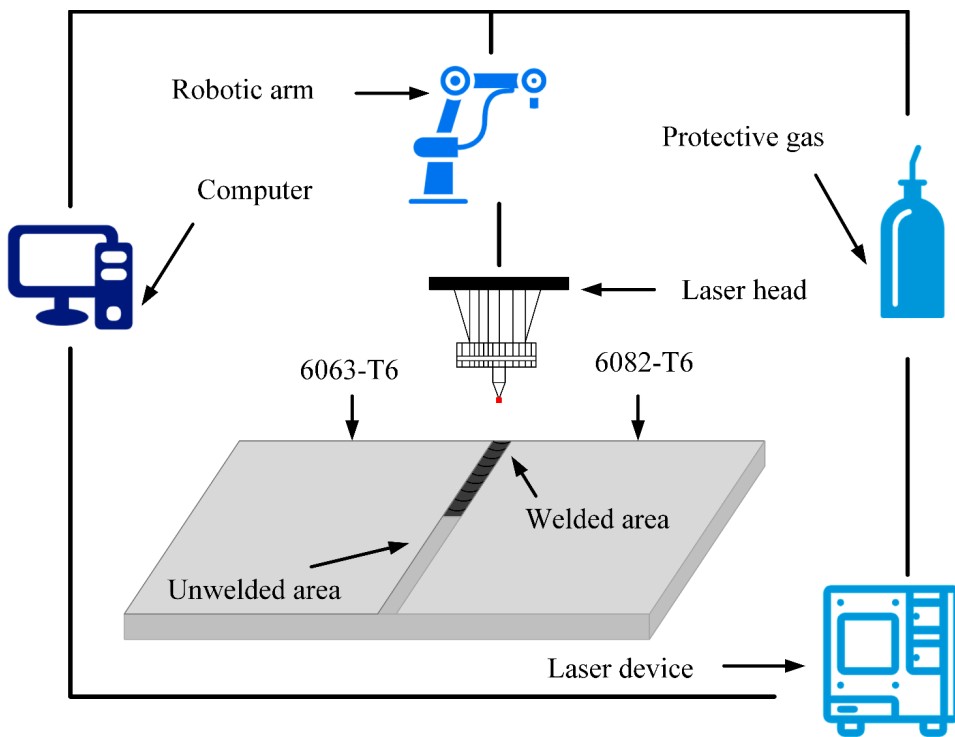

**Figure 1.** Schematic diagram of laser welding.

**Table 2.** Chemical composition of BM (wt.%).

| Materials | Si | Mn | Mg | Cu | Zn | Ti | Fe | Al |
|---|---|---|---|---|---|---|---|---|
| 6063-T6 | 0.381 | 0.009 | 0.707 | 0.040 | 0.003 | 0.029 | 0.244 | Bal. |
| 6082-T6 | 1.000 | 0.560 | 1.000 | 0.030 | 0.060 | 0.030 | 0.330 | Bal. |

*2.2. Experimental Methods*

After welding, samples for microstructure analysis and mechanical properties testing were extracted using a wire cutting machine (DK-7745, Xiongfeng, Taizhou, China). For surface analysis, different types of sandpaper (180#, 400#, 800#, 1000#, 1200#, 1500#, 2000#) were used to grind the surface longitudinally to horizontally alternately. Then a metallographic polishing machine (MP-2, Sanfeng, Guangzhou, China) was used to transverse polish the grinding surface in a clockwise direction and a polishing agent composed of diamond powder and grinding medium was used to improve the polishing quality. Finally, the polished surface was corroded according to the standard (1 mL HF + 200 mL $H_2O$) corrosion reagent [8]. The size of electron backscattered diffraction (EBSD) sample was 3 mm × 10 mm × 2 mm (SIGMA, ZEISS, Shanghai, China). After mechanical polishing, the sample was electropolished in 10% perchloric acid anhydrous ethanol solution. The temperature was −20 °C and the polishing time was 40 s. After calibration, the EBSD data were processed by the HKL Channel 5 commercial software package. The microhardness test was carried out (HV-1000, three tests, Guangzhou, China), and the distance between the test points was between 0.5–1.0 mm. The samples for tensile property testing were produced according to ASTM B557M-15 standard [9], the specific dimensions of the samples are shown in Figure 2. Tensile specimens were repeated twice (EUT5105, Sansi, Shenzhen, China), and the average value was taken for data processing. After tensile fracture, scanning electron microscopy (SEM) was used to evaluate the fracture morphology.

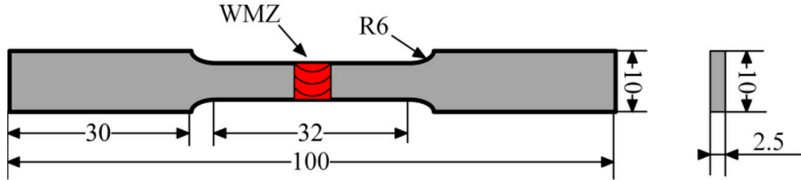

**Figure 2.** Tensile specimen dimensions.

## 3. Numerical Simulation

### 3.1. Mathematical Model

In order to balance computational efficiency with accuracy, a symmetric model was adopted. The dimensions of this model were 20 mm × 8 mm × 2.5 mm. The mesh was densified in the vicinity of the weld, and the upper layer of the workpiece included an air domain within the thickness range of 1 mm, as shown in Figure 3. The model simplification incorporated several assumptions [10–13], which included the following:

(1) The liquid metal in the welding pool was treated as an incompressible Newtonian fluid exhibiting laminar flow.

(2) The material properties, such as specific heat capacity, viscosity, and thermal conductivity, were temperature dependent. The material density was considered constant, and thermal buoyancy was modeled using the Boussinesq approximation.

(3) Evaporation and metal loss during the welding process were not factored into the model.

(4) The influence of protective gas was neglected.

(5) Table 3 lists the symbols for parameters used in the model. Table 4 lists the parameter symbols in numerical simulation models.

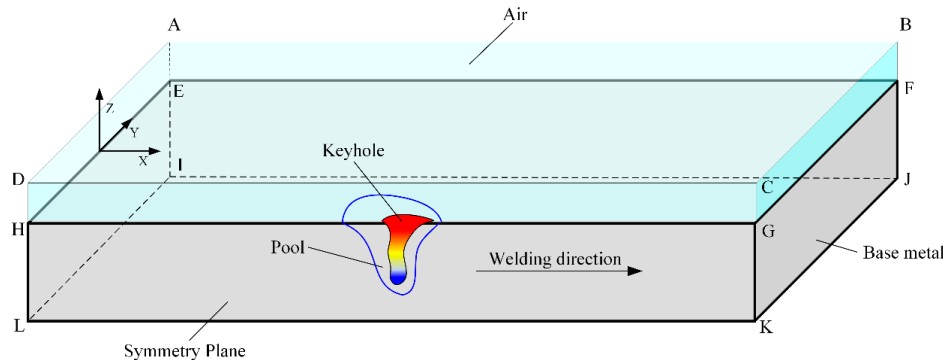

**Figure 3.** Schematic of the computational domain of the model.

**Table 3.** Parameter symbols in numerical simulation models.

| Symbol | Nomenclature | Symbol | Nomenclature |
| --- | --- | --- | --- |
| $t$ | Time (s) | $S_m$ $S_e$ | Source term |
| $f_L$ | Liquid fraction | $f_\infty$ | Ambient temperature (k) |
| $F$ | Volume fraction | $T_L$ | Solidus temperature (k) |
| $h_c$ | Convection heat transfer coefficient (W/m$^2$·K) | $T_s$ | Liquidus temperature (k) |
| $h$ | Heat source depth (m) | $\overrightarrow{v}$ | Velocity vector (m/s) |
| $H$ | Mixing enthalpy (J/kg) | $u_1$, $v_1$, $w_1$ | Velocity components (m/s) |
| $k$ | Thermal conductivity (W/m·K) | $\rho_1$ | Density (kg/m$^3$) |
| $\eta$ | Thermal efficiency | $\sigma$ | Stefan–Boltzmann constant (W/m$^2$·K$^4$) |
| $\overrightarrow{n}$ | The normal vector of the free surface | $\varepsilon$ | Emissivity |
| $P$ | Input power(W) | $\mu$ | Dynamic viscosity (kg/m·s) |
| $\delta$ | A minor constant | $A$, $B_o$ | Material-related constants |
| $Q$ | Heat flow density (W/m$^2$) | $U$ | Latent heat of evaporation (J/kg) |
| $R$ | Heat source radius(m) | | |

**Table 4.** Thermal properties of aluminum alloy.

| Parameters | Symbol | Unit | Value |
|---|---|---|---|
| Solidus temperature | $T_S$ | | 820 |
| Liquidus temperature | $T_L$ | K | 930 |
| Boiling point | $T_b$ | | 2730 |
| Solidus Density | $\rho_S$ | kg/m$^3$ | 2700 |
| Liquidus Density | $\rho_L$ | | 2400 |
| Solidus Specific heat | $C_S$ | J(kg·K) | 871 |
| Liquidus Specific heat | $C_L$ | | 1060 |
| Latent heat of fusion | $\Delta H_f$ | J/kg | $3.87 \times 10^5$ |
| Latent heat of evaporation | $\Delta H_v$ | | $1.08 \times 10^7$ |
| Solidus Thermal conductivity | $\lambda_S$ | W/(m·K) | 238 |
| Liquidus Thermal conductivity | $\lambda_L$ | | 100 |
| Surface tension | $\sigma$ | N/m | 0.914 |
| Surface tension gradient | $\frac{\partial_\sigma}{\partial_T}$ | N/(m·K) | $-0.35 \times 10^{-3}$ |
| Ideal gas constant | $R$ | J/(mol·K) | 8.314 |

*3.2. Governing Equations*

Adhering to the outlined assumptions, the governing equations for mass, momentum, and energy conservation were formulated alongside the volume of fluid (VOF) method. These equations were integral in simulating heat transfer, free surface dynamics, keyhole behavior, and flow within the molten pool [14–19].

The mass conservation equation is articulated as follows:

$$\frac{\partial \rho_1}{\partial t} + \nabla \rho \vec{v} = 0. \tag{1}$$

The energy conservation equation is expressed in the following form:

$$\frac{\partial(\rho H)}{\partial t} + \frac{\partial(\rho u_1 H)}{\partial x} + \frac{\partial(\rho v_1 H)}{\partial y} + \frac{\partial(\rho w_1 H)}{\partial z} = \nabla(K\nabla T) + S_E. \tag{2}$$

The expression of the momentum conservation equation is as follows:

$$\rho_1 \left( \frac{\partial \vec{v}}{\partial t} + \vec{v} \cdot \nabla \vec{v} \right) = -\nabla p + \mu A \frac{(1 - f_L)^2}{f_L^3 + \delta} + S_m. \tag{3}$$

If the liquid volume fraction in the solid–liquid paste region was set to change linearly with temperature, the following equations were satisfied:

$$f_L = \begin{cases} 0, & (T \leq T_s) \\ \frac{T - T_s}{T_s - T_L}, & (T_s \leq T \leq T_L). \\ 1, & (T \geq T_L) \end{cases} \tag{4}$$

The expression of the VOF equation was as follows:

$$\frac{\partial F}{\partial t} + \nabla \left( \vec{v} F \right) = 0. \tag{5}$$

*3.3. Boundary Conditions*

In the context of welding, heat transfer between the welding components and the weld primarily occurs through radiation, conduction, and convection. The workpiece

interface was treated as symmetrical, and the energy conservation equation was established as follows [20–22]:

$$k\frac{\partial T}{\partial \vec{n}} = Q - h_c(T - T_\infty) - \varepsilon\sigma\left(T^4 - T_\infty^4\right). \tag{6}$$

The tiny hole in the welding pool during the laser welding process was brought on by the recoil steam pressure that resulted from the workpiece material evaporating. This statement is as follows [23]:

$$P_r = 0.54\frac{AB_0}{\sqrt{T}}\exp\left(-\frac{U}{\sigma T}\right). \tag{7}$$

In the center of symmetry plane boundary conditions:

$$v = 0, \frac{\partial u}{\partial y} = 0, \frac{\partial v}{\partial y} = 0, \frac{\partial T}{\partial y} = 0. \tag{8}$$

The free interface boundary condition of the molten pool surface is as follows:

$$\mu\frac{\partial u}{\partial z} = \frac{\partial \gamma}{\partial x}, \mu\frac{\partial v}{\partial z} = \frac{\partial \gamma}{\partial y}. \tag{9}$$

At the solid–liquid phase interface:

$$u = 0, v = 0, \omega = 0. \tag{10}$$

### 3.4. Laser Heat Source Model

The heat flow density of laser welding diminishes along the thickness of the workpiece during the welding process. Taking into account the real-world welding scenario, employing a rotating surface Gaussian heat source as the welding heat source is deemed most appropriate [24]. Figure 4 illustrates the heat source model, and the heat source equation is expressed as

$$q(x,y,z) = \frac{9\eta P}{\pi h(1 - e^{-3})}\exp\left\{-9\frac{x^2 + y^2}{R^2\ln\left(\frac{h}{z}\right)}\right\}. \tag{11}$$

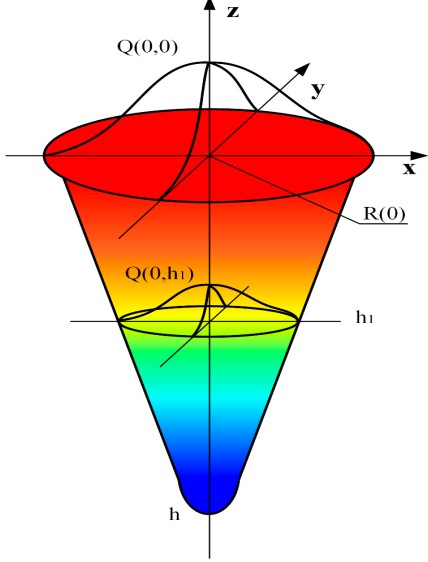

**Figure 4.** Schematic diagram of welding heat source.

## 4. Results and Discussion

### 4.1. Verification of Heat Flow Coupling Mathematical Model

This study focused on comparing the shape of the actual welding pool with the predictions of the numerical simulation to assess the model's accuracy and reliability. The welding process was simulated and experimentally tested under specific conditions of welding speed and laser power. Table 5 presents the comparison between the calculated results of the weld cross-section shape and the experimental findings under various laser power settings. The model's accuracy was calibrated and validated based on the fusion line's position in the weld section and the weld pool's width, as detailed in Table 6.

**Table 5.** Comparison between actual weld pool and simulated weld section under different laser power.

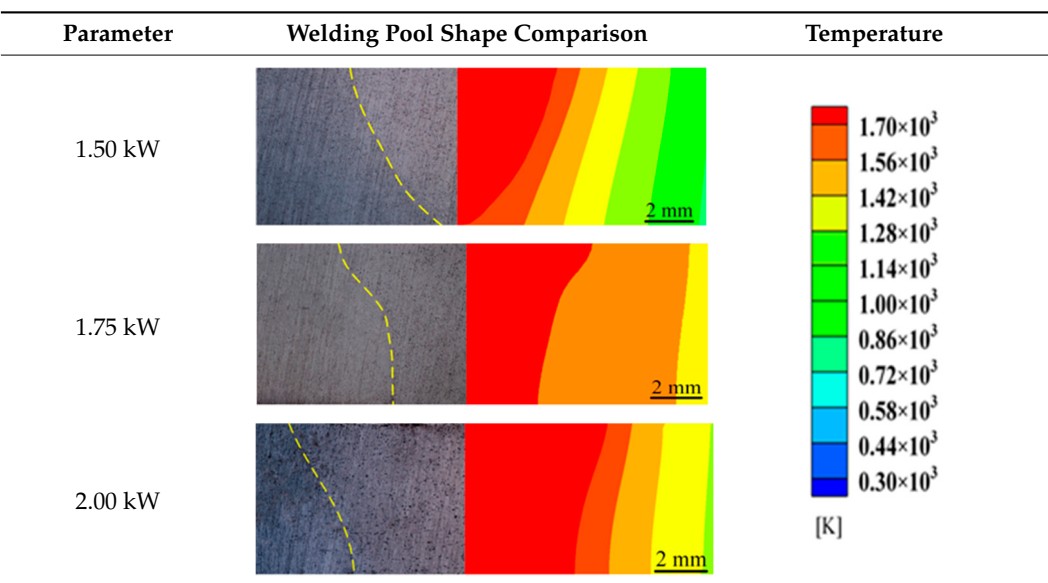

**Table 6.** Simulation and experimental measurement data of molten pool cross-section.

| Item | 1.50 kW | 1.75 kW | 2.00 kW |
|---|---|---|---|
| Simulation value of melting width (mm) | 4.3 | 9.7 | 11.2 |
| Measurement value of melting width (mm) | 4.5 | 9.8 | 11.5 |
| Rate of error (%) | 4.4 | 1.0 | 2.6 |

The comparative analysis of results across the three different welding process parameters revealed a strong congruence between the weld pool morphology calculated in the numerical simulation and the actual test outcomes. The discrepancies between the numerical simulation calculations and the actual measurements were found to be within a reasonable margin. Consequently, the mathematical model established in this research is deemed valid, and the selected Gaussian heat source model of the rotating surface accurately simulates the welding process.

### 4.2. Influence of Laser Power on Flow Behavior of Welding Pool

Laser power plays a pivotal role in the actual welding process significantly impacting weld formation and joint quality. Achieving an optimal balance is crucial, as insufficient laser power may result in incomplete welding penetration, while excessive power can cause spatter and porosity, adversely affecting the quality of the welded joint. Therefore, the flow characteristics of the welding pool under varying laser powers were examined by simulating the flow field at three different laser power levels.

The flow state of the weld pool is intrinsically linked to the driving forces within it. As depicted in Figure 5a, at a laser power of 1.5 kW, there were no small eddy currents observed behind the welding pool; rather, a large convection circulation was present. This effect was due to the relatively lower laser power producing a shallower melt depth and the surface area of the weld pool being larger than its depth, consequently augmenting the range of surface tension. In this scenario, the absence of significant metal vaporization and the low recoil pressure could not sustain themselves within the welding pool. Thus, under the dominance of surface tension, the welding pool exhibited flow characteristics similar to thermal conduction welding.

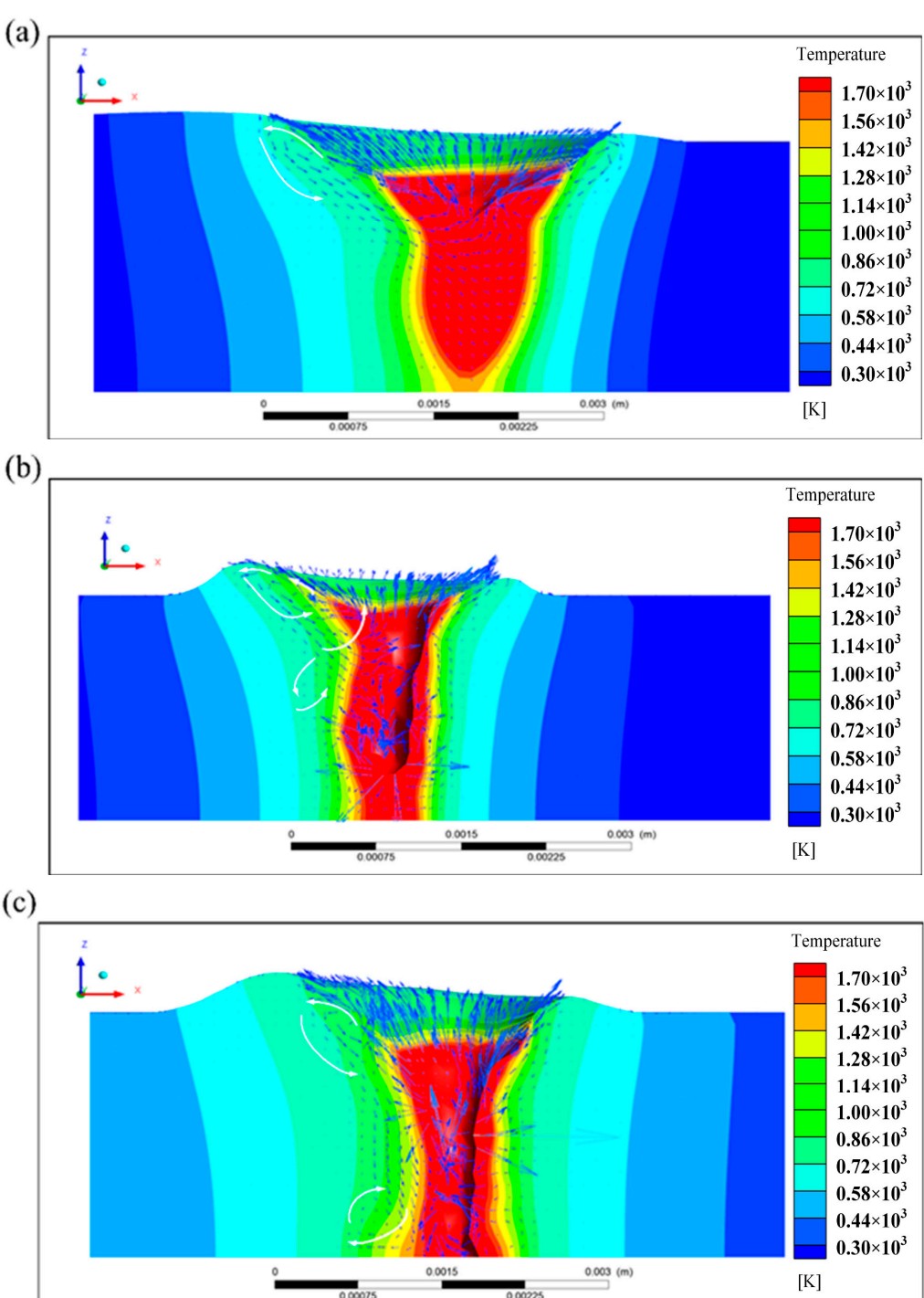

**Figure 5.** Influence of laser power on flow behavior of weld pool: (**a**) 1.50 kW; (**b**) 1.75 kW; (**c**) 2.00 kW.

When the laser power was increased to 1.75 kW, the high-energy laser beam caused strong vaporization of the welding substrate. This resulted in a gradual stabilization of the recoil pressure in the welding pool and a noticeable change in the flow of liquid metal near the rear. At this juncture, the forces within the welding pool predominantly comprised recoil pressure and surface tension, prompting an upward flow of the liquid metal. The effect of recoil pressure led to a flow in the liquid metal opposite to the convection ring behind the weld pool. Additionally, the increased laser power also intensified the eddy currents behind the weld pool, as illustrated in Figure 5b.

Increased laser power can significantly influence the stability of liquid metal flow in the welding pool. When the laser power was elevated to 2.00 kW, the convection ring between the upper and lower halves of the welding pool became more pronounced. While the metal flow direction above the welding pool remained largely unchanged, a distinct eddy current ring emerged below. This phenomenon can be attributed to the stabilization of the recoil pressure within the welding pool, reducing its impact and making surface tension the primary driver of the welding pool flow.

*4.3. Weld Morphology*

Figure 6 showcases the macroscopic morphology of the welds' upper and root surfaces at different laser powers. A well-formed structure was observed on both the upper and root surfaces of the welds, with laser power settings ranging from 1.50 to 2.00 kW. Notably, there were no welding flaws, such as faulty welds or collapses. With the increase in laser power, the weld's width on the upper surface expanded noticeably, a consequence of the increased welding heat input and the heat source's wider impact diameter, resulting in a broader weld. The root surface of the weld displayed a nonlinear change pattern under varying laser powers. At 1.50 kW and 2.00 kW, the weld width varied somewhat irregularly while maintaining a continuous morphology, as illustrated in Figure 6a,c. At a laser power of 1.75 kW, there was no significant change in the width of the root surface of the weld, as shown in Figure 6b. This consistency can be attributed to the welding process being smooth and there was no instability in the root shape of the weld caused by excessive or insufficient heat input. Optimal macroscopic morphology of the weld was achieved at a laser power of 1.75 kW.

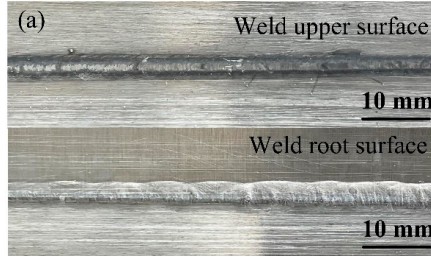 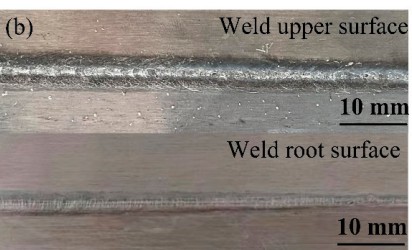 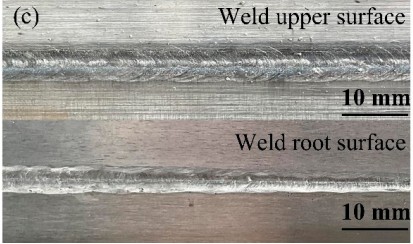

**Figure 6.** Macroscopic morphology of the weld under varying laser powers: (**a**) 1.50 kW; (**b**) 1.75 kW; (**c**) 2.00 kW.

*4.4. Microstructure of the Welded Joints*

The microstructure of the BM (6063-T6 and 6082-T6 aluminum alloys) is illustrated in Figure 7. The grains are uniformly oriented along the rolling direction, with the primary features being a dispersed Al matrix and coarse, black-dot reinforced phases. These black dotted phases, primarily $Mg_2Si$, are a result of silicon and magnesium being the main additives in both materials. A comparative analysis of the distribution and size of the reinforced phases in the two materials reveals that the 6082-T6 Aluminum Alloy the BM has widely distributed and coarse strengthening phases. As shown in Table 7, the grain size for the 6063-T6 aluminum alloy and the 6082-T6 aluminum alloy were measured at 18.85 μm and 15.56 μm, respectively.

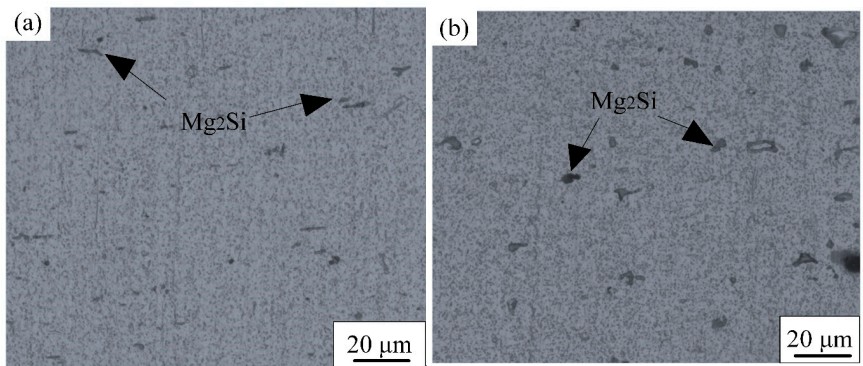

**Figure 7.** Microstructure of BM: (**a**) 6063-T6; (**b**) 6082-T6.

**Table 7.** Average grain size of each area of the WJ when the laser power was 1.75 kW.

| | Welded Joint | | | | |
|---|---|---|---|---|---|
| | **BM-6063** | **HAZ-6063** | **WMZ** | **HAZ-6082** | **BM-6082** |
| Grain size (μm) | 18.85 | 36.64 | 45.42 | 33.18 | 15.65 |

Figure 8 displays the microstructure of WJ with a laser power of 1.75 kW. The Euler diagram of the WJ, based on the results of the EBSD test, is depicted in Figure 9. The WMZ near the FL on both sides consisted mainly of columnar crystals, while equiaxed crystals predominated in the WMZ center. This variation is attributed to the relationship between the temperature gradient and solidification velocity ratio (R) in the main grain shape of the weld solidification structure. Larger R values favor the formation of columnar or planar grains, whereas smaller R values lead to predominantly equiaxed grains. The WMZ near the FL exhibited a higher heat dissipation coefficient and temperature gradient, coupled with a relatively lower solidification speed. Conversely, the WMZ center, filled with abundant liquid metal, had a lower heat dissipation capacity compared to the BM, resulting in a smaller R and consequently distinct crystal morphologies in the WMZ. The grain orientation map of the WJ at 1.75 kW laser power is presented in Figure 10. The orientations are indicated by color: red for <001>, blue for <111>, and green for <101>. The prevalence of blue and green grains near the welding fusion line on both sides suggests dominant grain orientations of <111> and <101>, indicating columnar crystal growth towards the maximum temperature gradient. The grain sizes in the HAZ near the 6063-T6 aluminum alloy side and the HAZ near 6082-T6 aluminum alloy sides were 36.64 μm and 33.18 μm, respectively. The average grain size in the WMZ was larger than that of the HAZ on both sides, and its grain size was 45.52 μm.

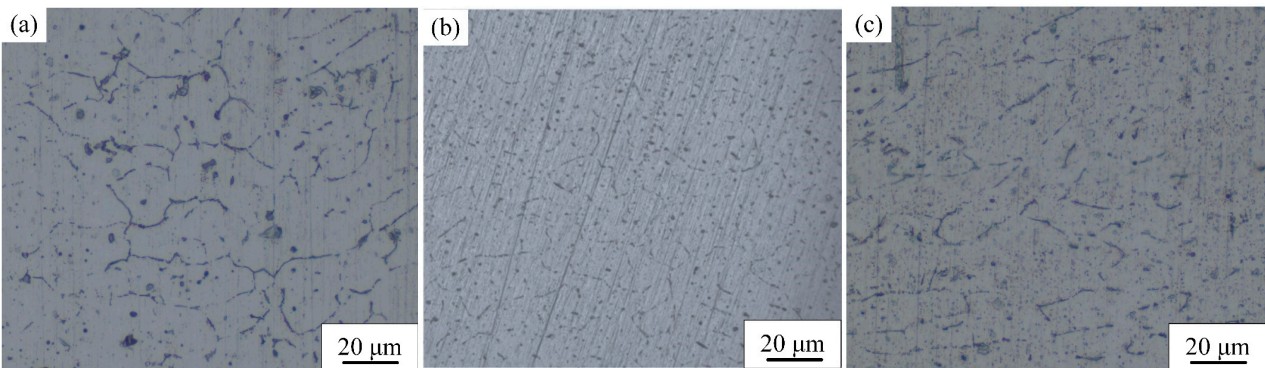

**Figure 8.** Microstructure of the WJ (laser power was 1.75 kW): (**a**) the 6082 side of the HAZ; (**b**) WMZ; (**c**) the 6063 side of the HAZ.

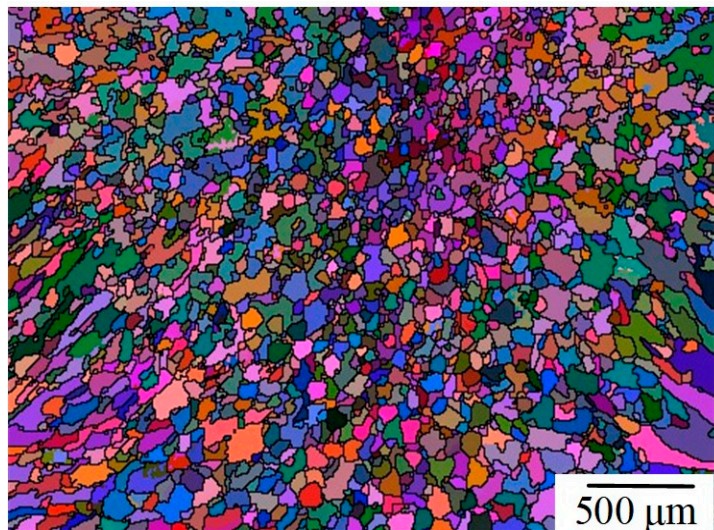

**Figure 9.** Euler diagram of the WJ when the laser power was 1.75 kW.

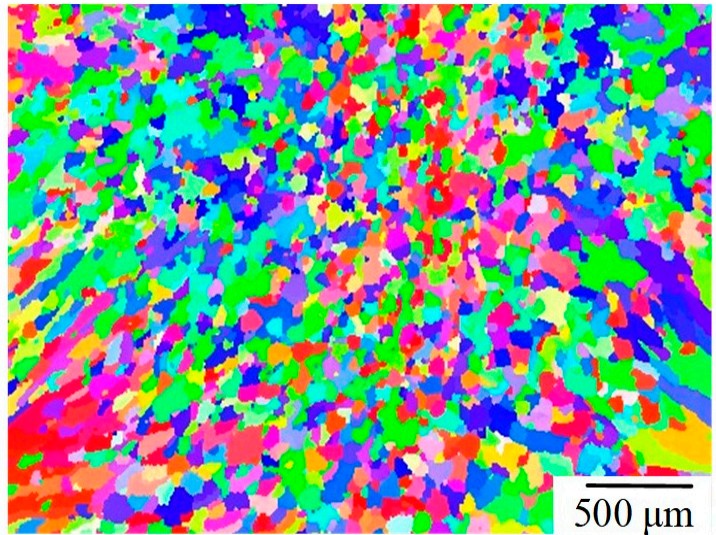

**Figure 10.** Grain orientation map of the WJ when the laser power was 1.75 kW.

*4.5. Mechanical Properties*

4.5.1. Microhardness

The Vickers microhardness of the welded joint was evaluated in the transverse direction at a laser power of 1.75 kW. The average microhardness values for the WMZ, HAZ, and BM are depicted in Figure 11. The 6082-T6 aluminum alloy exhibited an average microhardness of 107.8 HV with a deviation of 5.4. The 6063-T6 aluminum alloy had an average microhardness of 92.4 HV, with a deviation of 1.0. Notably, the microhardness in the WMZ was the lowest. The average microhardness and the error of the WMZ was 73.46 HV and 1.0, respectively. This was mainly because there were a lot of coarse columnar crystals in the WMZ near the FL on both sides. The grain sizes in the HAZs on both sides were smaller than those in the WMZ, resulting in the average microhardness of the HAZs being intermediate between that of the WMZ and the BM. The average microhardness of the HAZ on the 6082 side was higher than that on the 6063 side, measuring 79.94 HV and 75.98 HV, respectively.

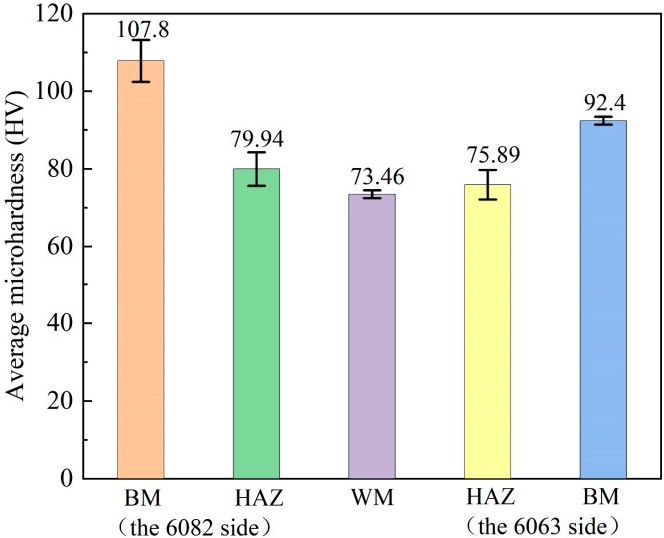

**Figure 11.** Average value of the microhardness of the WJ under laser power was 1.75 kW.

### 4.5.2. Tensile Properties

The tensile test results, including the stress–strain curves and tensile properties data, are presented in Figure 12 and Table 8. The highest tensile strength, recorded at 308.13 MPa, was observed in the 6082-T6 aluminum alloy, while the 6063-T6 aluminum alloy exhibited the greatest elongation, measuring 8.34%. At a laser power of 1.75 kW, the WJ demonstrated a tensile strength of 180.8 MPa and an elongation of 4.04%. Figure 13a illustrates that at this laser power, the tensile fracture occurred in the HAZ adjacent to the 6063-T6 aluminum alloy side. The tensile test results indicated that the HAZ near the 6063-T6 side was the weakest area in terms of tensile properties for the WJ welded at 1.75 kW was the HAZ near 6063-T6 aluminum alloy side. Despite the average grain size of WMZ being larger than that of the HAZ, the coarse columnar region was small. The welding thermal cycle induced residual stress in the HAZ, consequently diminishing its mechanical properties. Furthermore, the presence of the $Mg_2Si$ reinforced phase in the BM originally contributed to higher mechanical properties. During welding, the $Mg_2Si$ reinforced phase and the grain which absorb heat without melting in the HAZ can be heated and growth. Compared to their original size of $Mg_2Si$ reinforced phase, the strengthening effect of the larger size $Mg_2Si$ reinforced phase was weaker or vanishing, leading to decreased tensile properties of the WJ.

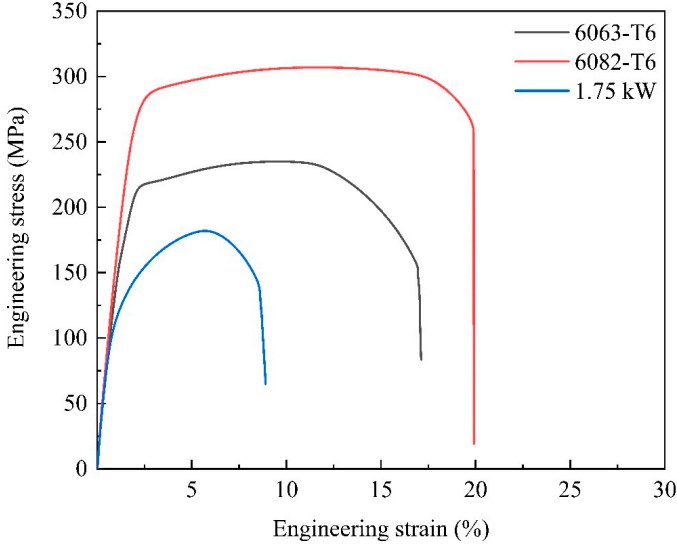

**Figure 12.** Tensile test curves.

**Table 8.** Tensile properties.

| Test | Samples | Tensile Strength (MPa) | | Elongation (%) | |
|---|---|---|---|---|---|
| | | Single Value | Average Value | Single Value | Average Value |
| 1 | 6063-1 | 235.01 | 237.64 | 9.33 | 8.34 |
| 2 | 6063-2 | 240.26 | | 7.34 | |
| 3 | 6082-1 | 306.99 | 308.13 | 8.62 | 8.17 |
| 4 | 6082-2 | 309.02 | | 7.71 | |
| 5 | 1.75-1 | 179.63 | 180.8 | 4.02 | 4.04 |
| 6 | 1.75-2 | 181.97 | | 4.06 | |

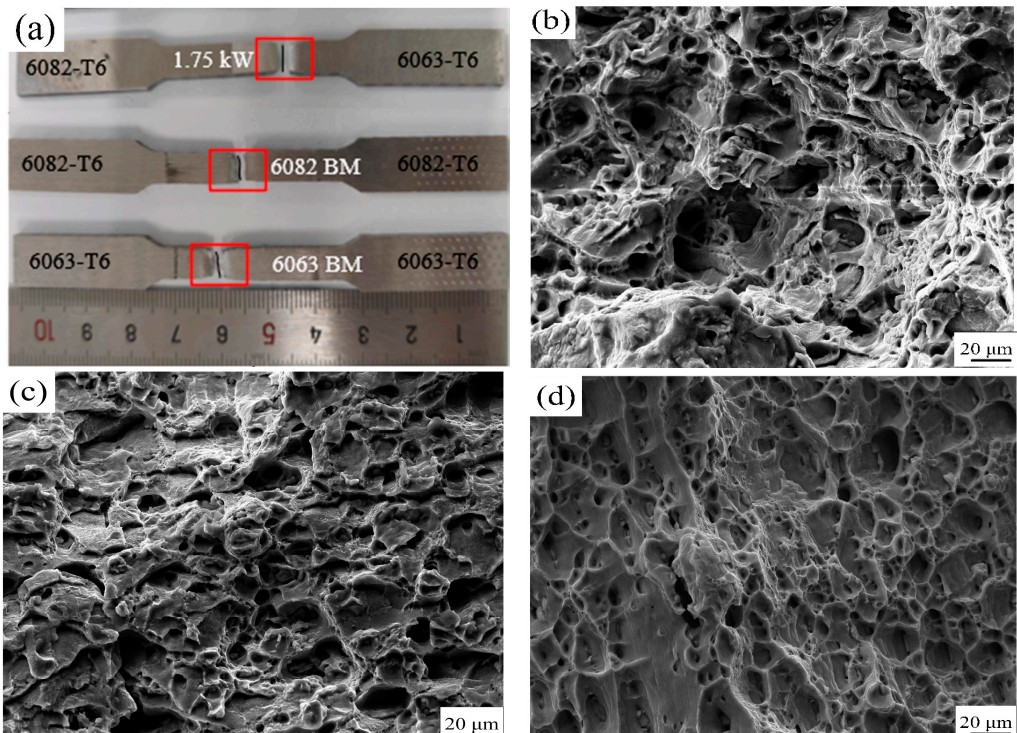

**Figure 13.** Specimens after tensile fracture: (**a**) tensile fracture location; (**b**) 6063-T6 fracture morphology; (**c**) 6082-T6 fracture morphology; (**d**) fracture morphology of welded joint with 1.75 kW laser power.

In the post-tensile test, the micromorphology of all tensile fractures is depicted in Figure 13b–d. The fracture surfaces of the 6063-T6 and the 6082-T6 aluminum alloys were characterized by dimples, indicative of ductile fractures, as shown in Figure 13b,c. Compared with the fracture of the 6063-T6 aluminum alloy, the dimple size of the 6082-T6 aluminum alloy tensile specimen was obviously smaller than that of the 6063-T6 aluminum alloy. At 1.75 kW laser power, the WJ's tensile fracture predominantly featured dominated by dimples. Compared with the 6063-T6 and 6082-T6 aluminum alloys, the dimples were shallower in depth and less in number per unit area. It has been provided that fine dimples can enhance tensile strength and microhardness under the effect of precipitation hardening. Additionally, the depth of the dimples is indicative of the material's deformation capability; greater plasticity is associated with deeper dimples. The fracture surface characteristics of the WJ correspond with the tensile property results mentioned above and further validate the conclusions drawn from the tensile tests.

## 5. Conclusions

(1) The accuracy of the rotary surface Gauss heat source in the numerical simulation was verified by comparing the weld pool morphology with the simulation results at welding powers of 1.5 kW, 1.75 kW, and 2.00 kW, respectively. The results showed that the weld pool morphology was in good agreement and the width error was less than 5%.

(2) With the increase in welding power, the stability of the molten pool decreased, and the liquid metal in the molten pool flowed backward and upward, mainly because the recoil pressure in the molten pool was stable, and the vortex ring was generated under the joint action of it and the surface tension, thus determining the flow form of the liquid metal in the molten pool.

(3) The macro morphology of the weld was greatest when the laser power was 1.75 kW. Columnar crystals made up the WMZ near the FL on both sides, while equiaxed crystals predominated in the WMZ center. The grain orientation in the WMZ near the FL on both sides were mainly <111> and <101>.

(4) At 1.75 kW laser power, WMZ had an average microhardness of 73.46 HV, lower than BM and HAZ. The tensile strength of 6063-T6 aluminum alloy side HAZ was the lowest, and the fracture mode was ductile fracture. The tensile strength was 180.8 MPa and the elongation was 4.04%.

(5) Although many scholars have conducted extensive research on aluminum alloy laser welding technology, their work has primarily been limited to macro- and micro-level studies of the same material after welding. This paper, however, explores the molten pool flow state, grain orientation, and mechanical properties of joints in heterogeneous aluminum alloy laser welding, providing more convincing research conclusions for the connection of body structural components and playing a guiding role. Future research can focus on the corrosion-susceptibility of 6063-T6 and 6082-T6 aluminum alloy joints and its impact on mechanical properties.

**Author Contributions:** Methodology, Y.Y.; Formal analysis, H.C.; Data curation, S.Z.; Writing—original draft, S.C. and F.T.; Writing—review & editing, S.C. All authors have read and agreed to the published version of the manuscript.

**Funding:** This work was financially supported from the China Postdoctoral Science Foundation under Grant (number 2021MD703809) and Science and Technology Planning Project of Liuzhou under Grant (number 2022JRZ0101) and Guangxi University of Science and Technology Doctoral Fund under Grant (number 19Z27).

**Institutional Review Board Statement:** Not applicable.

**Informed Consent Statement:** Not applicable.

**Data Availability Statement:** Data are contained within the article.

**Conflicts of Interest:** The authors declared that they do not have any commercial or associative interests that represent a conflict of interest in connection with the work submitted.

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
