# Peer review of "Numerical Simulation and Mechanical Properties of 6063/6082 Dissimilar Joints by Laser Welding"

_coatings, doi:10.3390/coatings13122049_

Round 1

Reviewer 1 Report

Comments and Suggestions for Authors

Manuscript is well written with planned experimental results and the comparison with simulated results are really impressive. However, this manuscript is not at all matching with the socpe of the submitted journal 'coatings'. I would strongly recommond to transfer or submit it relevant journal. However, a few of my comments or observations are given below: 

1) The experimental section should have all equipment or instruments used in the present study represented in a standard format (Model, make, city, and country)

2) Table 4: images should have a scale/micro bar to make it user-friendly and authenticate results

3) Why is there variation in grain size for different materials after welding? please bring the reason with evidence. 

Comments on the Quality of English Language

poor English language; please check the overall manuscript for grammar and spelling corrections: 

1) Page no: 2; line no: 41: it should be 'these' instead of 'they'

2) Authors list ended with 'and'?

3) check superscripts: for e.g.: page no: 4; table 2: heat flow density units

Reviewer 2 Report

Comments and Suggestions for Authors

The authors have conducted an interesting research in an important research area. However the article should be revised in the light of the following observations:

1. The authors have selected AA6063 and 6082 alloys. This selection should be justified by discussing the commercial and industrial importance of this alloy in terms of its share in the total tonnage of aluminium based alloy used in the industry as well as its tonnage in metal usage in the world industry. Similarly the selection of laser welding should be justified against other methods like EBW etc.

2. The authors have mentioned that that these alloys have low density and high strength. The properties of these alloys should be presented as numerical values in tabular for to cover density, hardness, strength, UTS, toughness, ductility, weldability etc.

3. The micrograph quality should be improved especially in table 4.

4. In section 2.2 complete and detailed procedure should be presented in tabular form in terms of the steps involved in the grinding/polishing, the agents used (grit/diamond/silica/alumina etc) and the equipment used. Also the time of each sequence and how was it done (force on sample, rotation speed of plate/head, direction CW/CCW) should be specified for the interest of prospective readers.

5. The authors mention that the macroscopic appearance of 1.75 kW sample was the best. This should be justified interms of quantifiable values for standard welding quality tests like dye penetrant test, ultrsonic/xray tests or any other NDT or destructive test specified in the relevant standards and past literature.

6. Conclusions section should be strengthened to highlight the novelty of this research.

Comments on the Quality of English Language

 formal technical reporting format should be strictly followed.

Reviewer 3 Report

Comments and Suggestions for Authors

Dear autors

This is an interesting manuscript that requires some improvement before being considered for publication.

--Improve organization and novel aspects.

--Need to improve abstract and introduction to discuss the general problem and then present the specifics of this particular case.   

 --It is not good to combine result and discussion chapters, separate and describe separately.

--Figure 2. Tensile specimens _added draw a weld. Was the weld on the sample longitudinal or transverse?

----present more detailed and better supported conclusions

--use more current (less than 5 yrs old) references.

Round 2

Reviewer 1 Report

Comments and Suggestions for Authors

The revised manuscript is acceptable for publication; hence may please be accepted.

Author Response

Thank you for approving the manuscript!

Reviewer 2 Report

Comments and Suggestions for Authors

While all other observations have been addressed, response to the observation 1 " industrial importance of AA6063 and 6082 alloys in terms of its share in the total tonnage of aluminium based alloy used in the industry as well as its tonnage in metal usage in the world industry" is inadequate. What is the % use of the alloys AA6063 and 6082 out of the total usage of aluminium based alloys in the world?

Comments on the Quality of English Language

Minor proof reading is required.

Reviewer 3 Report

Comments and Suggestions for Authors

My opinion is that despite the authors made some improvements, the manuscript needs add discussion: Authors should discuss the results and how they can be interpreted in perspective of previous studies and of the working hypotheses. The findings and their implications should be discussed in the broadest context possible and limitations of the work highlighted. Future research directions may also be mentioned.
